# Training with Growing Sets:
# A Simple Alternative to Curriculum Learning and Self Paced Learning

## Abstract

Curriculum learning and Self paced learning are popular topics in the machine learning that suggest to put the training samples in order by considering their difficulty levels. Studies in these topics show that starting with a small training set and adding new samples according to difficulty levels improves the learning performance. In this paper we experimented that we can also obtain good results by adding the samples randomly without a meaningful order. We compared our method with classical training, Curriculum learning, Self paced learning and their reverse ordered versions. Results of the statistical tests show that the proposed method is better than classical method and similar with the others. These results point a new training regime that removes the process of difficulty level determination in Curriculum and Self paced learning and as successful as these methods.

## 1 Introduction

Bengio et al. (2009) named Curriculum learning the idea of following an order related with difficulty of the samples during training which provides an optimization for non convex objectives. After this many researchers tried to find the most efficient curriculum to get the best yield with this approach. In Zaremba & Sutskever (2014)'s study conventional curriculum learning did not work so well and they developed a new version. Shi et al. (2015) proposed three different curriculum strategies for language model adaptation of recurrent neural networks. In the field of computer vision Pentina et al. (2015) looked for the best order of tasks to learn. Although these models have better generalization performance with the proposed curriculum methods it is not known whether the tried methods ensures the best curriculum.

A curriculum is work-specific so could not be applicable for another work. In order to use the curriculum logic in different applications Kumar et al. (2010) suggested a method that the learner decides itself which samples are easy or difficult at every stage. This method called Self paced learning was combined with Curriculum learning which provides prior information by Jiang et al. (2015). In another work Graves et al. (2017) introduced a method to automatically select the syllabus to follow for the neural networks. Matiisen et al. (2017) also proposed a way to learn simple subtasks before the complex tasks and achieved better results than using manually designed curriculum.

In some cases higher learning performance could be obtainable by adding some noises to easy-to-hard ordering of the samples. Jiang et al. (2014) gave preference to both easy and diverse samples and outperform the conventional Self paced learning (Kumar et al. (2010)) algorithm. Emphasizing the uncertain samples suggested by Chang et al. (2017) lead to more accurate and robust SGD training. Avramova (2015) explored the inversed versions of the Self paced learning and Self paced learning with diversity (Jiang et al. (2014)) and demonstrated that these methods performed slightly better than their standard variants. Consistent with the literature we have showed in our previous work () that using both curriculum and anti-curriculum strategies improving generalization performance in a wide application area. These researches brings a question to minds: While it is natural and logical to obtain better results by sorting the samples from easy-to-hard why it is also better to sort the samples from hard-to-easy?

In this study we point that to start with a small training set and add new samples in both curriculum and anti-curriculum learning makes these methods better. So we claim that it is possible to have

Table 1: Training sets at each stage of CL

| Stage | Training set |
|---|---|
| $1^{st}$ stage | $1^{st}$ group |
| $2^{nd}$ stage | $1^{st}$ group + $2^{nd}$ group |
| ... | $1^{st}$ group + $2^{nd}$ group + ... |
| $n^{th}$ stage | $1^{st}$ group + $2^{nd}$ group + ... + $n^{th}$ group |

better results only by adding new samples stage-by-stage without a meaningful order. We experimented two ordering types related with difficulty (easy-to-hard and hard-to-easy) and our method without a meaningful order. Training was carried out by adding a new group to the training set at every stage. We compared the proposed method with two strategies. First one is Curriculum learning which we give the difficulty levels of the samples as pre-information. Second one is Self paced learning which the trained network determines the difficulty levels of the samples at each stage. All methods including usual baseline training have been compared by using paired T-test and the results are examined.

## 2  METHODS

We explored 2 ordering types determined by 2 different strategies and our method without ordering.

### 2.1  CURRICULUM LEARNING

In the Curriculum learning(CL) learner follows a pre-determined curriculum during training. For this reason it is necessary to know the difficulty of the samples. This information can be given by defining the specific curriculum of the task or labeling the samples at the beginning. Even though labeling with difficulty levels is easy for artificial data sets, it is costly and demanding for real world data sets. Besides that, a sample which is easy for humans may not be easy for machines. In this study, we used an ensemble method to automatically determine the difficulty of the samples. We created an ensemble with Bagging method(Breiman (1996)) and calculated decision consistency for each training sample accordingly the difficulty levels are determined. Then, the ordered training set which is easy-to-hard was grouped for Curriculum learning and hard-to-easy ordered training set was grouped for Anti-curriculum learning(ACL).

For CL, the samples are sorted firstly then separated into same sized groups by dividing the number of training samples into the number of stages. Remainings are added to the first stage so that the number of new samples to add at each stage is equal. Training starts from the easiest group and continues by adding the groups one-by-one according to the easiness. Hardest group is given with the entire training set at the last stage and training is completed in stages same number as groups.

For ACL, the training set is sorted by difficulty in descending order and then separated into groups. Training starts with the group that is the most difficult. At the last stage of the training, the group includes the easiest samples with the entire training set. The groups given at each stage are shown in Table 1 when the training set considered as sorted according to the difficulty levels.

At each step a new group of samples are given together with the previous samples for both methods. In the last stage the entire training set is given like the baseline training. The path for CL is given in Algorithm 1. The same algorithm is applied for ACL with reverse order training set.

Curriculum learning starts with random initialized weights in the first stage. The optimum weights found at each stage taken as the initial weights of the next stage. In other words, next stage of the optimization starts from the minimum found in the previous stage.

### 2.2  SELF PACED LEARNING

Self paced learning(SPL) is a solution for finding the difficulty levels in Curriculum learning. In this method learner determines the instances to learn according to the current situation of the objective

---

**Algorithm 1** Curriculum learning

---
 1: $D \leftarrow$ training set sorted by easiness
 2: $n \leftarrow$ number of stages
 3: $s \leftarrow$ number of samples to add at each stage
 4: $f \leftarrow$ non-convex objective function of neural network with parameters $\theta$
 5: **procedure** CL$(D, n, s, f)$
 6: Randomly initialize the parameters $\theta_0$
 7: $D_0$ = first s sample of D
 8: $\theta_1 = argmin_\theta f(D_0, \theta_0)$
 9:     **for** $t = 1$ **to** $n - 1$ **do**
10:     $D_t$ = first $(t + 1) * s$ sample of D
11:     $\theta_{t+1} = argmin_\theta f(D_t, \theta_t)$
12:     **end for**
13:     **return** $\theta_n$
14: **end procedure**

---

function. Training starts with a group of random samples and the samples that best fit the current model space are labeled as easy at each step. Weights are updated by easy labeled examples at the next step. The network is trained with more examples in the next step and all training set is given in the final step. This is realized by annealing the self pace parameter. Same number of samples at each stage can be added to the training set to make the training progress homogeneous.

In this work the only parameter to set for SPL is the number of stages. The number of samples to add at each stage is equal like as CL. Training starts with randomly selected samples. At the end of each stage, the samples are sorted by ascending order according to the losses calculated with MSE. The samples to give in the next stage are selected from this order. The most consistent samples with the model space are selected for the next stage.

For the Self paced learning-Inversed(SPLI), the most outlying samples have priority. Same as SPL, training starts with random samples. Training samples in the next stage are selected by considering the incompatibility with the model space. All training set is sorted by loss in descending order, samples with the highest loss are selected for the next stage. The path for SPL is given in Algorithm 2. The same algorithm is applied for SPLI, only training set is sorted by reverse order at each stage.

Training in the next stage is started with using previous solutions in SPL also as in CL. SPL does not guarantee that the all samples from previous stage will be taken again in the next stage. The samples best adapted with the objective function are taken among all training samples and some of the samples used in the previous stage may not be inside these samples. For example, if there is noisy samples in the first stage these probably will not be in the training set at the second stage.

---

**Algorithm 2** Self paced learning

---
 1: $T \leftarrow$ random ordered training set
 2: $n \leftarrow$ number of stages
 3: $s \leftarrow$ number of samples to add at each stage
 4: $f \leftarrow$ non-convex objective function of neural network with parameters $\theta$
 5: **procedure** SPL$(T, n, s, f)$
 6: Randomly initialize the parameters $\theta_0$
 7: $T_0$ = randomly chosen s sample from T
 8: $\theta_1 = argmin_\theta f(T_0, \theta_0)$
 9:     **for** $t = 1$ **to** $n - 1$ **do**
10:     $D_t$ = training set sorted by loss of $f(T, \theta_t)$ ascending order
11:     $T_t$ = first $(t + 1) * s$ sample of $D_t$
12:     $\theta_{t+1} = argmin_\theta f(T_t, \theta_t)$
13:     **end for**
14:     **return** $\theta_n$
15: **end procedure**

---

### 2.3 Random ordered growing sets

CL, SPL and their inverse versions have a common point: At each next step the number of training samples is increasing with addition of a new group. Due to the both versions (easy-to-hard and hard-to-easy) have better performance than the classical method it is considered that this common feature may provide an optimization. We investigated the total weight change in each iteration for all methods in the Appendix A. In this case, it is important to train with accumulating groups rather than giving the samples with a meaningful order. Therefore, dividing the unordered training set into the groups and adding a new group at each stage could also raise the generalization performance.

In the Random ordered growing sets(ROGS) method unsorted training set is divided into groups with equal number of samples. Training is carried out in the same fashion as starting with the first group then adding a new group at each stage without a meaningful order. Algorithm 3 shows the way of training with Random ordered growing sets. Same as the other methods, ROGS uses the found solutions in the previous stages as initial weights in the next stage.

---

**Algorithm 3** Random ordered growing sets

---

1: $T \leftarrow$ random ordered training set
2: $n \leftarrow$ number of stages
3: $s \leftarrow$ number of samples to add at each stage
4: $f \leftarrow$ non-convex objective function of neural network with parameters $\theta$
5: **procedure** ROGS($T, n, s, f$)
6: Randomly initialize the parameters $\theta_0$
7: $T_0 =$ randomly chosen s sample from T
8: $\theta_1 = argmin_\theta f(T_0, \theta_0)$
9:     **for** $t = 1$ **to** $n - 1$ **do**
10:     $T_t =$ randomly choose $(t + 1) * s$ sample from T
11:     $\theta_{t+1} = argmin_\theta f(T_t, \theta_t)$
12:     **end for**
13:     **return** $\theta_n$
14: **end procedure**

---

Examined 5 methods find the optimum weights with the same objective:

$$\theta_{t+1} = \underset{\theta \in R^d}{argmin} \sum_{i=1}^{(t+1)*s} (f(x_i, \theta_t) - y_i)^2 \qquad (1)$$

where $s$ denotes the number of samples to add at each stage, parameters at each stage determined by same amount of samples for all methods. $x_i, y_i$ are taken from pre-sorted easy-to-hard training set for CL, from pre-sorted hard-to-easy training set for ACL, from the training set that sorted easy-to-hard at each stage for SPL and from the training set that sorted hard-to-easy at each stage for SPLI. In ROGS method samples are selected from non-ordered training set.

## 3 A theoretical explanation about training with growing sets

**Definitions: (a)** The loss function $\ell(\theta)$ with one parameter $\theta \in R$ is given as in (2) where N is the number of training instances and the loss function for each individual instance $\ell_i(\theta)$ is given as in (3).

$$\ell(\theta) = \frac{1}{N} \sum_{i=1}^{N} (f(x_i, \theta) - y_i)^2 \qquad (2)$$

$$\ell_i(\theta) = (f(x_i, \theta) - y_i)^2 \qquad (3)$$

**(b)** The point $\theta_B$ is a definite local minimum of $\ell(\theta)$. Geometric representation of a definite local minimum point is shown in Figure 1(a) with the loss functions. We show the loss functions for 10

instances with thin lines and their average with the bold line. If we denote the expected value of $\ell(\theta)$ as $E[\ell(\theta)]$, expected values of the loss function in the given points can be ordered as in (4).

$$E[\ell(\theta_A)] > E[\ell(\theta_C)] > E[\ell(\theta_B)] > E[\ell(\theta_D)] \tag{4}$$

Expected values of the derivatives of the loss function in the given points can be written as in (5) and the derivatives are given in Figure 1(b).

$$\ell'(\theta_A) = E[\frac{1}{N}\sum_{i=1}^{N}\ell_i'(\theta_A)] < 0$$

$$\ell'(\theta_B) = E[\frac{1}{N}\sum_{i=1}^{N}\ell_i'(\theta_B)] = 0$$

$$\ell'(\theta_C) = E[\frac{1}{N}\sum_{i=1}^{N}\ell_i'(\theta_C)] = 0 \tag{5}$$

$$\ell'(\theta_D) = E[\frac{1}{N}\sum_{i=1}^{N}\ell_i'(\theta_D)] < 0$$

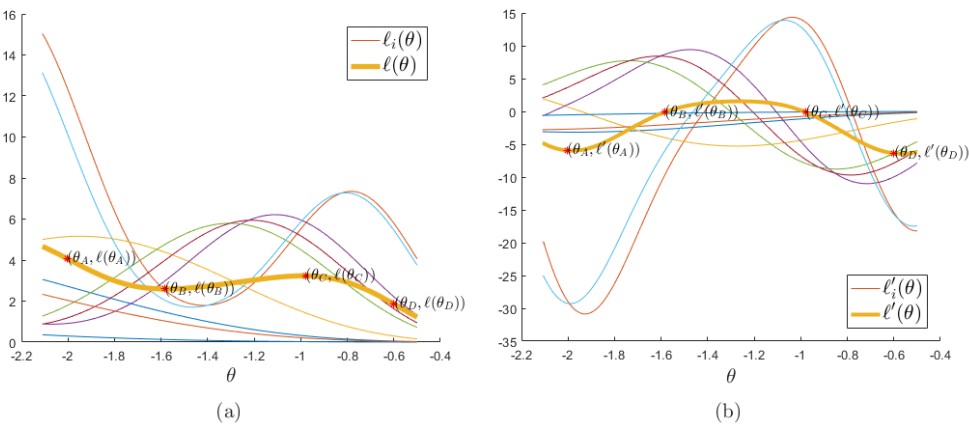

Figure 1: Loss functions of individual instances vs. all instances(a) and derivatives(b)

**(c)** We define the complexity as the number of critical points and denote with $\kappa$. Complexity of the $\ell(\theta)$ is $\kappa(\ell(\theta))$.

**Assumptions: (a)** Complexity of the loss functions of each individual instance can be as much as the complexity of the loss function of all instances.

$$\kappa[\ell_i(\theta)] <= \kappa[\ell(\theta)] \qquad \forall i \in \{1, 2, ..., N\} \tag{6}$$

**(b)** More than half of the individual instances are less complex than average of all instances.

$$\kappa[\ell_i(\theta)] < \kappa[\ell(\theta)] \qquad (m > \frac{N}{2}) \tag{7}$$

**Lemma 1:** Probability density function of the derivatives in the $\theta_B$ point has a skewed distribution with zero mean.
*Proof:* Derivatives of the loss functions is under the point $\theta_B$ in Figure 1(b) for more than half of the individual instances. However $\ell_i'(\theta_B)$ values are high for the above instances. If we denote the probability density function of the derivatives in the $\theta_B$ as $Pr(\ell_i'(\theta_B))$,

*1)* Definition (c) gives that $\theta_B$ is a local minimum therefore $Pr(\ell'_i(\theta_B))$ has zero mean.
*2)* If $\kappa(\ell(\theta)) = 2$ as in Figure 1(b) then $\kappa(\ell_i(\theta)) <= 2$ from Assumption (a) and $\kappa(\ell_i(\theta)) < 2$ for more than half of the instances from Assumption (b). $\ell'_i(\theta_B) < 0$ for the instances that has $\kappa(\ell_i(\theta)) < 2$. $\ell'_i(\theta_B) < 0$ for more than half of the instances therefore $Pr(\ell'_i(\theta_B))$ has a skewed distribution.

**Corollary 1:** If we take a subset with *k* instances from the training set we can denote:

$$\ell_s(\theta_B) = \frac{1}{k} \sum_{i=1}^{k} \ell_i(\theta_B) \tag{8}$$

Expected value for the derivative of the loss function of the subset at the point $\theta_B$ is less than zero.

$$E[\ell'_s(\theta_B)] < 0 \tag{9}$$

*Proof:* Probability of being negative for $E[\ell'_s(\theta_B)]$ depends on $k$ and always high from Lemma 1.

**Theorem 1:** If we take a subset with *k* instances and train with batch gradient descent, we obtain a better local minimum for this subset.
*Proof:* Optimization in the $\ell(\theta)$ surface which starts from $\theta_A$ will probably end at $\theta_B$. However in the $\ell_s(\theta)$ surface optimization will be continue to find a better minimum for the current set according to Corollary 1.

**Theorem 2:** To continue the optimization with all samples provides a better local minimum for all training set than stopping at the minimum of the subset.
*Proof:* Error surface for all instances ($\ell(\theta)$) and some subsample instances ($\ell_{sx}(\theta), \ell_{sy}(\theta), \ell_{sz}(\theta), \ell_{sw}(\theta)$) are given in Figure 2. These examples shows all possible situations for the stopping points by considering $\theta_A, \theta_B$ and $\theta_C$.

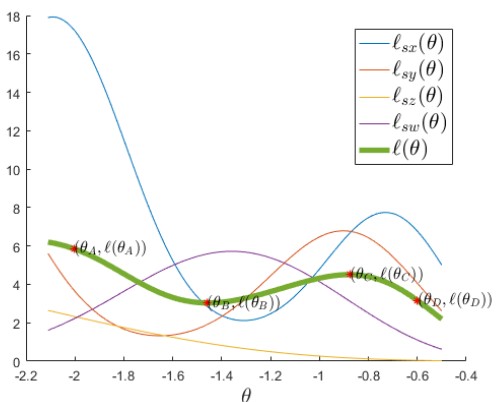

Figure 2: Loss functions of subsample instances vs. all instances

We suppose to start the optimization from $\theta_A$.
- If we use $\ell(\theta)$ it will be stop at $\theta_B$.
- If we use $\ell_{sx}(\theta)$ it will be stop at $\theta_{sx}$ in the range $\theta_B < \theta_{sx} < \theta_C$. If we finish the optimization here, error on the surface for all instances will be high ($E[\ell(\theta_{sx})] > E[\ell(\theta_B)]$). Therefore optimizing in the $\ell(\theta)$ subsequently prevents to stay at a worse point.
- If we use $\ell(\theta_{sy})$ it will be stop at $\theta_{sy}$ in the range $\theta_A < \theta_{sy} < \theta_B$. Here $E[\ell(\theta_{sy})] > E[\ell(\theta_B)]$ therefore optimization for all instances is necessary.
- If we use $\ell(\theta_{sz})$ it will be stop at $\theta_{sz}$ in the range $\theta_C < \theta_{sz}$ and it is possible to obtain a better local minimum with optimization on $\ell(\theta)$ surface by starting from the local minimum of the subset.
- If we use $\ell(\theta_{sw})$ it will be stop at $\theta_{sw}$ in the range $\theta_A > \theta_{sw}$. Here the stopping point can be worse for all instances. Therefore it is possible to avoid from worse point by optimization on the $\ell(\theta)$ surface.

Table 2: T-test results

|         | Baseline | CL      | ACL     | SPL     | SPLI    | ROGS    |
|---------|----------|---------|---------|---------|---------|---------|
| **Baseline** | -        | 0/24/12 | 6/17/13 | 0/29/7  | 3/13/20 | 3/16/17 |
| **CL**       | 12/24/0  | -       | 11/20/5 | 4/32/0  | 4/20/12 | 7/24/5  |
| **ACL**      | 13/17/6  | 5/20/11 | -       | 7/18/11 | 0/24/12 | 1/30/5  |
| **SPL**      | 7/29/0   | 0/32/4  | 11/18/7 | -       | 3/18/15 | 5/24/7  |
| **SPLI**     | 20/13/3  | 12/20/4 | 12/24/0 | 15/18/3 | -       | 7/29/0  |
| **ROGS**     | 17/16/3  | 5/24/7  | 5/30/1  | 7/24/5  | 0/29/7  | -       |

**Theorem 3:** We can obtain a better local minimum with training with growing sets.
*Proof:* Subset of the training set provides to get a better local minimum as in Theorem 1. By the same way subset of the subset surface can be provide a better local minimum for the subset. To optimize the surface of the whole training set first optimize the following surfaces respectively:

$$\ell_{S_1}(\theta), \; \ell_{S_2}(\theta), \; ..., \; \ell(\theta) \qquad s(S_1) < s(S_2) < ... < N \tag{10}$$

## 4 IMPLEMENTATION

We trained 3-layer artificial neural network which has 10 neurons in the hidden layer with stochastic gradient descent with momentum for all methods. Stop condition is the rising validation error series for 6 times during the training. Baseline and all other methods applied with incremental training. We set the number of stages as 25 for all methods with growing sets. The network must provide the stopping condition at each stage to pass the next stage. Optimized weights in the previous stage are given as initial weights in the next stage.

We have tested the suggested methods in 36 data sets retrieved from UCI repository[1] with the sample number range from 57 to 20000. We divide each data set 5 folds, 3 for training, 1 for validation and 1 for testing at each experiment. For each data set 20 error rates (MSE) obtained with 4x5 fold cross validation are compared by 0.95 significance level paired T-test. Results of the comparisons are in Table 2. (Abbreviations are as follows: Curriculum learning=CL, Anti-curriculum learning=ACL, Self paced Learning=SPL, Self paced learning-Inversed=SPLI, Random ordered growing sets=ROGS) Each cell contains the win/tie/loss information for corresponding row against corresponding column. For example, Curriculum learning wins versus Baseline in 12 data sets, ties in 24 data sets, not loses in any data set. Results of the comparisons with Baseline for all data sets are given in the Appendix B with the number of samples, number of features, number of classes and average Baseline MSE. If the comparison result of the corresponding method against Baseline is win it is marked with W, tie marked with T and loss marked with L.

It is seen that training with ROGS is better than Baseline method in 17 data sets. Considering the comparisons with the Baseline, ROGS method wins in more data sets than CL, ACL and SPL. Obtaining good results without a meaningful order shows that only giving the training set as growing subsets provides an optimization. Additionally, SPLI method wins against Baseline in more than half of the data sets. This method seems as the best method in terms of total number of wins.

It is striking that CL and SPL methods did not lose against Baseline in any data set. This shows the robustness of these methods in noisy data sets. It is possible to give priority to noises with hard-to-easy and random ordered methods so training may be misdirected in the data sets with high error rates. In such data sets giving priority to easy examples provides a safer training.

## 5 DISCUSSION AND FUTURE WORK

We drew our attention that both versions of training with easy-to-hard ordered and hard-to easy ordered samples have better performance. That led us to investigate what common issues they have. We considered that their common point is growing the training sets during training. Therefore, instead of ordering the samples according to difficulty we only added some samples randomly at

---

[1] http://archive.ics.uci.edu/ml/datasets

each stage. In these experiments we obtained similar results with Curriculum, Anti-curriculum, Self Paced and Self Paced-Inversed methods which are related to difficulty levels. According to these results, we can claim that the success of Curriculum learning and Self paced learning approaches not comes from the fact that they follow a meaningful order but trained by growing training sets.

In Figure 1(a) we showed some examples for the individual instances. We started the optimization from $\theta_A$, instances under this point are considered as easy and above are difficult. If we take an easy instance it is possible or not to guide the optimization to a better minimum. It will be stop at the local minimum $\theta_B$ in the worst case. Similarly if we take a difficult instance it is possible or not to obtain a better result. Implementation results also showed that both easy-to-hard and hard-to-easy ordered methods can be successful. Therefore ordering of the samples are not so important to guide the optimization.

It is a better situation to shorten the distance between $\theta_B$ and $\theta_C$ in Figure 2 to bypass the local minimum. When the points are same for a saddle point, training with growing sets will probably overcome this point and find a better minimum. This is a good condition when considering saddle points are so much than local minimums in high dimensional functions as mentioned in Dauphin et al. (2014).

On many data sets with different distributions we used ensemble method to automatically determine the difficulty of the samples for curriculum learning. Pre-processing for difficulty level determination can be thought to caused slowdown. However it has provided a faster neural network training than SPL. Also it could be said that ensemble method set a better ordering than SPL by considering their number of wins against Baseline.

ACL and SPLI, which are the inverse versions of the CL and SPL methods, has performed poorly in some high error rated data sets. The effect of giving the samples at different points during the training has been studied in Erhan et al. (2010). In these methods, noisy examples may be effecting the output more because of giving at the beginning. Nevertheless, the inverse versions of the approaches have better performance than their standard versions. However, CL and SPL methods did not lose in any data set so this shows they have a robust aspect. It is thought that these methods must have a theoretical explanation about ensuring resistance to noises. Gong et al. (2016) studied on why these methods are effectiveness especially on big and noisy data.

SPLI method has the most winning against Baseline. In this method strategy of selecting the samples to learn at each stage reminds pool-based active learning(Lewis & Gale (1994)) in which the learner wants to learn the uncertain samples of the unlabeled data pool. Also non-loss of CL and SPL, and more wins of ACL and SPLI shows the necessity of determining the valuable-example-based curriculum instead of easiness-based-curriculum for the future work.

Our contributions:

- Previous work on CL and SPL has made experiments on specific domains. In this study, the performance of these methods on various data sets has been shown.

- We proposed a new method that need not to determine the difficulty levels as in CL and SPL, so faster and as successful as these methods.

- We explained the theoretical perspective of the training with growing sets.

- We recommend to use the ROGS method as a benchmark in subsequent CL and SPL studies. Only if there is a success up beyond the growing training sets, it can be said that this is the effect of the right ordering.

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

## A    EXPLORING THE TOTAL WEIGHT CHANGES AT EACH ITERATION

In order to find the common feature of CL, SPL and their reverse versions, we examined the total weight change in each iteration during training. Graphs of each method for vehicle data set are given in the Figure 3. In these graphs, total weight change is increasing and decreasing during stages. This may be indicates a feature that provides avoidance from the local minimum. By adding a new group at each stage, it is possible to continue with a larger step from the minimum of the previous stage.

Iterations of the last stage which we give the all training set is marked with orange. When the whole training set is given it reaches the minimum in a shorter time than baseline in all methods. This implies the whole training set has been started from a better minimum in the methods with growing sets than random initializing in the baseline.

The common feature of the methods are not related to the ordering of the samples. Only the growth of the training set at each stage with the addition of a new group made to continue the optimization with avoiding local minimum. In this case training with growing sets without ordering can be also provide a better minimum.

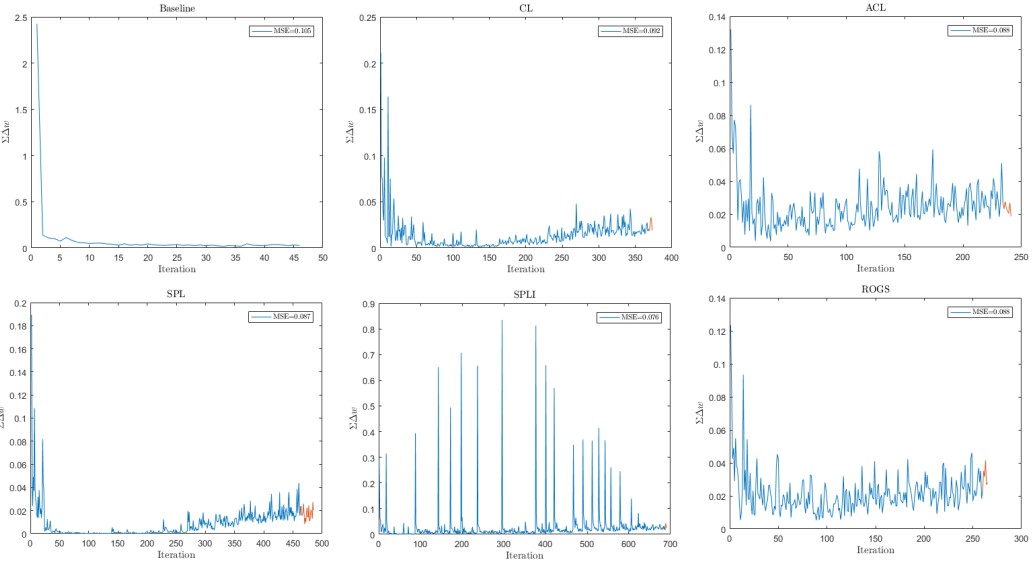

Figure 3: Total weight changes at each iteration

## B    COMPARISONS WITH BASELINE

| ID | Data set | # of Samp. | # of Feat. | # of Clas. | Avg. MSE | CL | ACL | SPL | SPLI | ROGS |
|---|---|---|---|---|---|---|---|---|---|---|
| 1 | labor | 57 | 26 | 2 | 0,15 | W | T | W | W | W |
| 2 | zoo | 84 | 16 | 4 | 0,02 | T | T | T | W | T |
| 3 | lymph | 142 | 37 | 2 | 0,14 | T | T | T | T | T |
| 4 | iris | 150 | 4 | 3 | 0,03 | T | W | T | W | W |
| 5 | hepatitis | 155 | 19 | 2 | 0,15 | T | T | T | T | T |
| 6 | audiology | 169 | 69 | 5 | 0,08 | T | W | T | W | W |
| 7 | autos | 202 | 71 | 5 | 0,13 | W | W | T | W | W |
| 8 | glass | 205 | 9 | 5 | 0,12 | T | T | T | T | T |
| 9 | sonar | 208 | 61 | 2 | 0,17 | T | T | T | W | T |
| 10 | heart-statlog | 270 | 13 | 2 | 0,14 | T | L | T | T | T |
| 11 | breast-cancer | 286 | 38 | 2 | 0,21 | T | L | T | L | L |
| 12 | primary-tumor | 302 | 23 | 11 | 0,08 | W | T | W | W | W |
| 13 | ionosphere | 351 | 33 | 2 | 0,10 | T | T | T | T | T |
| 14 | colic | 368 | 60 | 2 | 0,15 | T | L | T | L | L |
| 15 | vote | 435 | 16 | 2 | 0,04 | W | T | T | T | T |
| 16 | balance-scale | 625 | 4 | 3 | 0,06 | T | T | T | W | T |
| 17 | soybean | 675 | 83 | 18 | 0,03 | W | T | W | W | W |
| 18 | credit-a | 690 | 42 | 2 | 0,12 | T | T | T | T | T |
| 19 | breast-w | 699 | 9 | 2 | 0,03 | T | T | T | T | T |
| 20 | diabetes | 768 | 8 | 2 | 0,16 | T | T | T | T | T |
| 21 | vehicle | 846 | 18 | 4 | 0,09 | W | W | W | W | W |
| 22 | anneal | 890 | 62 | 4 | 0,01 | T | W | T | W | W |
| 23 | vowel | 990 | 11 | 11 | 0,06 | W | W | W | W | W |
| 24 | credit-g | 1000 | 59 | 2 | 0,20 | T | L | T | L | L |
| 25 | col10 | 2019 | 7 | 10 | 0,05 | W | T | T | W | W |
| 26 | segment | 2310 | 18 | 7 | 0,02 | W | W | T | W | W |
| 27 | splice | 3190 | 287 | 3 | 0,03 | T | L | T | T | T |
| 28 | kr-vs-kp | 3199 | 39 | 2 | 0,02 | T | W | T | W | W |
| 29 | hypothyroid | 3770 | 31 | 3 | 0,04 | T | W | T | T | W |
| 30 | sick | 3772 | 31 | 2 | 0,04 | T | W | T | W | W |
| 31 | abalone | 4153 | 10 | 19 | 0,04 | W | W | W | W | W |
| 32 | waveform | 5000 | 40 | 3 | 0,07 | T | L | T | T | T |
| 33 | d159 | 7182 | 32 | 2 | 3e-5 | W | W | W | W | W |
| 34 | ringnorm | 7400 | 20 | 2 | 0,10 | T | T | T | T | T |
| 35 | mushroom | 8124 | 112 | 2 | 3e-5 | T | W | T | W | T |
| 36 | letter | 20000 | 16 | 26 | 0,03 | W | T | T | W | W |

