# OpenReview forum: "Training with Growing Sets: A Simple Alternative to Curriculum Learning and Self Paced Learning"
_ICLR.cc/2018/Conference — Reject_

### Official Review · AnonReviewer2 · 2017-11-27
**This paper is trying to analyse different learning strategies, curriculum learning versus learning in random order, arguing that the latter one can achieve the same competitive performance as the former one.  The proposed approach of learning with growing sets as well as empirical results are not convincing.**

**Rating:** 4
**Confidence:** 4

**Review:**

This paper addresses an interesting problem of curriculum/self-paced versus random order of samples for faster learning. Specifically, the authors argue that adding samples in random order is as beneficial as adding them with some curriculum strategy, i.e. from easiest to hardest, or reverse.
The main learning strategy considered in this work is learning with growing sets, i.e. at each next stage a new portion of samples is added to the current available training set. At the last stage, all training samples are considered. The classifier is re-learned on each stage, where optimized weights in the previous stage are given as initial weights in the next stage.

The work has several flaws.
-First of all, it is not surprising that learning with more training samples at each next stage (growing sets) gets better - this is the basic principle of learning. The question is how fast the current classifier converges to the optimal Bayes level when using Curriculum strategy versus Random strategy. The empirical evaluations do not show evidence/disprove regarding this matter. For example, it could happen that the classifier converges to the optimal on the first stage already, so there is no difference when training in random versus curriculum order with growing sets.
-Secondly, easyness/hardness of the samples are defined w.r.t. some pre-trained (external) ensemble method. It is not clear how this definition of easiness/hardness translates when training the 3-layer neural network (final classifier). For example, it could well happen that all the samples are equally easy for training the final classifier, so the curriculum order would be the same as random order. In the original work on self-paced learning, Kumar et al (2010), easiness of the samples is re-computed on each stage of the classifier learning.
-The empirical evaluations are not clear. Just showing the wins across datasets without actual performance is not convincing (Table 2).
-I wonder whether the section with theoretical explanation is needed. What is the main advantage of learning with growing sets (when re-training the classifier)  and (traditional) learning when using the whole training dataset (last stage, in this work)?

---

> ### Author Response · Authors · 2017-12-13
> **Re:**
>
> Thank you for your review, we have some comments:
>
> -Firstly our paper does not include a speed test for methods to find the faster one. We compared the error rates of the methods and get better results than standard incremental training in many cases. We looked for the reason why CL, SPL and reverse ordered variants have better performance and found that their common property is training with growing sets. So we growed the sets without demanding difficulty level determination process and also get better results.
>
> -In the CL we ordered the training set according to prediction confidence of the ensemble(all samples can't have same confidence degree) and divided the ordered training set into n(=25) parts. First part includes 1/n of the training set. In the ROGS, random ordered training set is divided into n parts and 1/n is taken in the first stage. So it is not so possible to have the same set on the first stage of CL and ROGS. When we are growing the sets in both methods we get lower error rates in the following stages logically. We could consider to add our paper how the test set error changes during stages. Additionally, we implemented the original work of SPL (Kumar et al., 2010), determined the difficulty levels at the end of each stage and took the samples from this ordering.
>
> -We thought a table with actual performances will not be easy to read but we are considering to give the table of errors in the Appendix.
>
> -Finally we point Section 3 of our paper about theoretical perspective. We make an explanation about training with small sets in the previous stages provides a better starting point for the bigger ones.

---

### Official Review · AnonReviewer1 · 2017-11-28
**The paper proposes to study the influence of ordering in the Curriculum and Self paced learning. The paper is mainly based on empirical justification and observation. The results on 36 data sets show that to some extent the ordering of the training instances in the Curriculum and Self paced learning is not important. The paper involves some interesting ideas and experimental results. I still have some comments.**

**Rating:** 6
**Confidence:** 3

**Review:**

The paper proposes to study the influence of ordering in the Curriculum and Self paced learning. The paper is mainly based on empirical justification and observation. The results on 36 data sets show that to some extent the ordering of the training instances in the Curriculum and Self paced learning is not important. The paper involves some interesting ideas and experimental results. I still have some comments.

1.	The empirical results show that different orderings still have different impact for data sets. How to adaptively select an appropriate ordering for given data set?
2.	The empirical results show that some ordering has negative impact. How to avoid the negative impact? This question is not answered in the paper.
3.	The ROGS is still clearly inferior to SPLI. It seems that such an observation does not strongly support the claim that ‘random is good enough’.

---

> ### Author Response · Authors · 2017-12-13
> **Re:**
>
> Thank you, we have some comments about your reviews.
>
> 1. There are many studies about to find the optimum curriculum for specific datasets. Here we propose a general idea that obtains good results on many cases with growing sets. It is a different exciting research area to find the rules about 'when to use which ordering'. As a result of our research we have seen that our method and reverse order versions of CL, SPL can obtain better results in specifically multiclass(more than 3 class) datasets.
>
> 2. According to our results we have negative impact on datasets which has 3 or less classes and high error rated. Noisy inputs of these datasets may be chosen previously and minimum of these data is not a proper starting point for the rest of the data.
>
> 3. ROGS method removes the problem of difficulty level determination and obtains good results in many cases. May be each dataset have a proper ordering but there is also a common point of all successful methods and this comes from growing the training sets stage-by-stage, and start each stage from the end point of the previous stage.

---

### Official Review · AnonReviewer3 · 2017-12-01
**No new idea with inconclusive experiments**

**Rating:** 4
**Confidence:** 4

**Review:**

Summary:
The paper proposes an algorithm to do incremental learning, by successively growing the training set in phases. However as opposed to training using curriculum learning or self paced learning, the authors propose to simply add training samples without any order to their "complexity". The authors claim that their approach, which is called ROGS, is better than the classical method and comparable to curriculum/self paced learning. The experiments are conducted on the UCI dataset with mixed results.

Review:
My overall assessment of the paper is that it is extremely weak, both in terms of the novelty of method proposed, its impact, and the results of the experiments. Successively increasing the training set size in an arbitrary order is the first thing that one would try when learning incrementally. Furthermore, the paper does not clearly explain what does it mean by a method to "win" or "lose". Is some training algorithm A a winner over some training algorithm B, if A reaches the same accuracy as B in lesser number of epochs? In such a case, how do we decide on what accuracy is the upper bound. Also, do we tune the hyper-parameters of the model along the way? There are so many variables to account for here, which the paper completely ignores.

Furthermore, even under the limited set of experiments the authors conducted, the results are highly inconclusive. While the authors test their proposed methodology on 36 UCI datasets, there is no clear indication whether the proposed approach has any superiority over the previous proposed ones, such as, CL and SPLI.

Given the above weaknesses of the paper i think the impact of this research is extremely marginal.

The paper is generally well written and easy to understand. There are some minor issues though. For example, I think Assumption (a) is quite strong and may not necessary hold in many cases.

---

> ### Author Response · Authors · 2017-12-13
> **Re:**
>
> Thank you for your review, we want to explain some points:
>
> When learning incrementally in standard method, we give the whole training set sample-by-sample and it continues learning the same samples in the following epochs until convergence. When we use growing sets, first we give only one part of the training set sample-by-sample, find the minimum of this part then continue with first and second part of the training set in the second stage. We conclude that minimum of the previous part is a better starting point for next part.
>
> As we explain in our paper we get 20 error rates(MSE) with 4x5 fold cross validation for each data set in all compared methods and made 0.95 significance level paired T-test. If one method has statistically significant better results according to T-test it wins, if it has worse results it losses.
>
> When we are working with 36 datasets it is difficult to find the best hyper-parameters in neural network for each dataset. We use the same model for all datasets and show that ROGS method works on many cases whether the model is proper for the data or not.
>
> We have seen that reverse order versions of CL and SPL are good in related works and thought that it may not necessary to order the samples. Superiority is to obtain near results without ordering and thus we can throw off the complexity (easiness/hardness) determination process.
>
> About Assumption (a) the average of simple functions may be more complex or the average of complex functions may be simpler. It is difficult to say which of these is more possible without making a presupposion over the functions. Assumption (a) indicates the condition when the most of the function derivatives (ℓi’ (θB)) are smaller than 0 at θB. This means the probability of being less than 0 is higher for the sum of the derivatives of a randomly selected part. Even if Assumption (a) is not true, there is still possibility of being less than 0 for the sum of the derivatives of a randomly chosen part. So, it is possible to overcome the local minimum at θB with a random selection. The assumption indicates a situation where this possibility is higher. If the sum of the derivatives of a randomly selected part is greater than 0, this selection may lead the optimization to the wrong direction. But subsequent steps can still orient the optimization the correct direction as we have explained in Theorem 2.

---

### Decision · Program_Chairs · 2018-01-29
**ICLR 2018 Conference Acceptance Decision**

**Decision:**

Reject

**Comment:**

The authors give evidence that is certain cases, the ordering of sample inclusion in a curriculum is not important.  However, the reviewers believe the experiments are inconclusive, both in the sense that as reported, they do not demonstrate the authors' hypothesis, and that they may leave out many relevant factors of variation (such as hyper-parameter tuning).